# Development of the MSKP index: Risk model of musculoskeletal pain in Colombian adolescents

**Margareth Lorena Alfonso-Mora**[1], **María Alejandra Sánchez-Vera**[1],
**Miguel Angel Uribe-Laverde**[2]\*, **Andrea Milena García Becerra**[3], **Paola Sarmiento Gonzalez**[4]

**1** Physiotherapy program, Faculty of Nursing and Rehabilitation, Universidad de La Sabana, Campus Puente del Común, Chía, Colombia, **2** Grupo de Investigación en Física y Matemáticas Aplicadas, Faculty of Engineering, Universidad de La Sabana, Campus Puente del Común, Chía, Cundinamarca, Colombia, **3** Psychology program, Faculty of Behavioral Sciences, Universidad de La Sabana, Campus Puente del Común, Chía, Colombia, **4** Nursing program, Faculty of Nursing and Rehabilitation, Universidad de La Sabana, Campus Puente del Común, Chía, Colombia

\* miguelurla@unisabana.edu.co

## Abstract

### Objective

This study aimed to develop a model to evaluate the risk of musculoskeletal pain (MSKP) in adolescents and identify the associated factors.

### Methods

A total of 680 adolescents were surveyed, with assessments of chronic neck, back, and shoulder pain, and related factors such as mobile dependence, physical activity, sleep quality, and sociodemographics. A multivariate logistic regression model was employed, with feature selection through correlation analysis and Lasso regression, to identify significant predictors and establish a risk scale. The model's performance was evaluated using the area under the ROC curve.

### Results

The prevalence of positive MSKP cases was 22.6% [95% CI: 19.7% – 25.9%]. Despite the self-reporting nature of the input data, the model achieved an AUC-ROC of 0.82, demonstrating good discriminatory ability. Key predictors include sleep disturbances, high mobile dependency, engagement in household chores, age 16–18, and urban residence. Girls exhibited a higher propensity for MSKP. Engagement in football was the only feature associated with a reduction of positive MSKP probability. A risk model is proposed to group the students into tertiles with low (2.3%), medium (17.4%) and high (44.7%) prevalence of MSKP.

**Data availability statement:** All data files are available from the Harvard Dataverse database (https://doi.org/doi:10.7910/DVN/6NTV2M).

**Funding:** This study was funded by the University of La Sabana (2020 – Convocatoria Interna Para La Financiación De Proyectos De Investigación, Creación, Desarrollo Tecnológico E Innovación). Project Id: ENF-59-2020.

**Competing interests:** The authors have declared that no competing interests exist.

## Conclusions

The MSKP Index effectively stratifies adolescent risk based on key factors, with significant associations between MSKP and sleep problems, mobile dependency, age, and gender. Regular physical activity, especially soccer, emerges as a protective factor, supporting targeted prevention strategies.

## Introduction

Musculoskeletal pain stands out as one of the most common symptoms prompting numerous medical consultations during childhood and adolescence [1]. The origins of this condition are multifaceted, encompassing factors like posture-related issues stemming from sedentary habits, psychosocial considerations, repetitive activities, traumas, and even more intricate conditions such as juvenile arthritis and neurological disorders [2]. As time progresses, these discomforts may evolve into chronic conditions in adulthood, significantly diminishing overall quality of life.

The most reported cause of musculoskeletal pain in children and adolescents is sedentary behavior, which is a highly prevalent risk factor [3,4]. Sedentary behavior is largely attributed to the widespread use of electronic devices, which, while offering a wide range of beneficial applications for entertainment, such as learning and task optimization, also carry risks of musculoskeletal and neurocognitive disorders [5]. Recent research has established a direct connection between electronic device use and musculoskeletal pain in children and adolescents [6–8]. It has been observed that approximately 60% of adolescents who adopt a sedentary lifestyle have a high probability of experiencing musculoskeletal discomfort, such as pain in the neck, shoulders, elbows, or lower back [9].

Musculoskeletal disorders include "text neck syndrome," which occurs due to the forward tilt of the head when using mobile devices while writing text messages or browsing the Internet social networks [10]. This action considerably shifts the center of gravity towards the front of the body, which significantly increases the load supported by the neck muscles, as well as the ligaments, tendons, and vertebrae. Consequently, incorrect postural adaptations occur and may contribute to musculoskeletal pain in this region. In addition, other musculoskeletal symptoms are observed, such as stiffness in the hypothenar region of the hand, pain in the thumb or in the front part of the forearm, which are a result of repetitive movement of the fingers of the hand, especially the thumb. This action over time can lead to the development of diseases such as Quervain's tendonitis [11].

The use of screens and mobile devices can have a significant impact on the quality of sleep of children and young people. A recent study has revealed that exposure to screens just before going to bed is associated with a decrease in sleep quality, which, in turn, is related to physical exhaustion and the development of obesity problems in children. Similarly, a direct connection between reduced sleep time and screen exposure has been shown [12]. The overuse of screens among adolescents,

particularly before bedtime, correlates with an abnormal sleep cycle, resulting in insomnia and excessive daytime drowsiness [13].

Predictive models play a critical role in identifying risk factors and formulating preventive strategies for symptoms. Currently, various investigations are being carried out to anticipate musculoskeletal diagnoses in adults. A systematic review suggests the consideration of specific clinical tests, a thorough evaluation, and an exhaustive examination, as well as the inclusion of risk factors of a psychosocial nature [14].

O'Sullivan et al. [15] emphasize that lumbar pain in adolescents is not solely explained by physical factors such as posture or scoliosis, but also involves psychosocial variables like negative beliefs, poor mental health, and stress. Adolescent females are more prone to developing disabling pain, with potential contributions from genetic and environmental factors. Additionally, Hestbaek et al. [16] explore the concept of comorbidity in adolescents, suggesting that conditions such as headaches and abdominal pain may serve as predictors for lumbar pain in adulthood, supporting the hypothesis of shared underlying physiological mechanisms. Regarding the child population, predictive models of pain have identified the prolonged use of mobile devices for more than an hour, maintained postures, and levels of physical activity as significant predictors of musculoskeletal pain within the school demographic [17–19].

Excessive screen exposure reduces physical activity and disrupts circadian rhythms, leading to poor sleep quality. This creates a vicious cycle: decreased rest results in higher daytime fatigue, which lowers the energy and motivation to engage in movement, further promoting sedentary behaviors associated with screen use. This cycle not only contributes to the development of musculoskeletal pain but also impairs overall well-being and recovery, highlighting the importance of addressing screen habits in prevention strategies [20].

Based on the existing literature, we hypothesized that higher levels of mobile dependency and poorer sleep quality would be significant predictors of MSKP in adolescents. We also considered sociodemographic factors such as age, gender, and residential area as potential moderators. This study aimed to propose a model to predict the risk of occurrence of musculoskeletal pain (MSKP) in a Colombian sample of children and adolescents and determine the associated factors.

## Methods

### Study design

Based on a self-administered online questionnaire, a cross-sectional study was conducted on students from years 5th to 11th in educational institutions in a municipality in the Central Region of Sabana Centro, Colombia South America during September and October 2022, no seasonal effects on pain incidence are attributed to Colombia's location, as it is situated in a tropical zone where seasonal variations do not significantly influence environmental factors related to pain.

### Ethics statement

This research was carried out in two educational institutions. Ethical approval was received by the ethics committee of the University of La Sabana (Minute 021−18 Nov 2020). Prior to participation, all students provided informed assent, and their parents/guardians signed informed consent forms.

### Sample size

The sample size was determined utilizing Epidat 3.1, a program designed to estimate proportions for epidemiological analysis of tabulated data. This calculation considered the following parameters: a population size of 39,640 students, an anticipated proportion of 50%, a confidence level of 95%, and a precision of 5%. Consequently, a total of 680 participants were involved. An intentional sampling was used; the research team invited all students, and only those who agreed to participate were included. We note that the use of a 50% proportion in a priori sample calculations yields the largest required sample size and thus provides a conservative estimation.

## Participants

A total of 1,183 students were initially invited to participate, with informed consent successfully obtained from 685 individuals. The inclusion criteria comprised students of both genders, aged between 10 and 18 years, enrolled in years 5th to 11th. Students with physical or cognitive disabilities and those with pre-existing medical diagnoses, such as juvenile diabetes, were excluded. Upon scrutiny of the criteria, 5 students were excluded due to incomplete information. Consequently, the final sample consisted of 680 participants.

## Variables and instruments

A web-based questionnaire, comprising two sections, was designed. The first section investigated the sociodemographic characteristics of the participants: age, sex, school grade, living and school area, and method of travel to school. The second section included a Mobile Dependency Test (TDM), and a questionnaire on physical activity, musculoskeletal pain, and sleep quality. A copy of the questionnaire is available in Spanish (S1 Questionnaire) and English (S2 Questionnaire).

**Mobile dependence.** Mobile phone dependence was assessed through a short version of the Mobile Dependency Test (TDM) developed by Chóliz et al. 2016 with adolescent population from six regions of the world [21]. This test consists of 22 items scored on a scale of Likert from 0 to 4. The test score ranges between 0 and 88 points; the higher the score, the greater the dependency. The Cronbach's alpha for the TDM was calculated as 0.89, indicating a high level of reliability.

**Physical activity.** The IPAQ-Short, short for the International Physical Activity Questionnaire, assesses engagement in moderate to vigorous activities across various domains such as leisure time, occupational, domestic, and transportation. It uses the "last 7 days" as a reference period. Adolescents were asked to recall the type of activity, frequency (days per week), and duration (hours and minutes per day) for each activity performed during the past week. The PAQ-A has shown Cronbach's alpha coefficients that vary depending on the study and sample. In Spanish adolescents, values of $\alpha = 0.65$ and $\alpha = 0.67$ have been reported for test and retest, respectively, and $\alpha = 0.74$ [22].

**Sleep quality.** The Sleep Disturbances Scale for School-age Children questionnaire was used, which considers six factors related to sleep quality: 1) difficulty initiating sleep, 2) nightmares, 3) nocturnal awakenings, 4) somnambulism, 5) tiredness and difficulty waking up, and 6) daytime sleepiness. Each factor is evaluated from zero days until seven days. The Cronbach's alpha coefficient for the total scale of the Sleep Disturbance Scale for School-Aged Children (SDSC) has been reported between 0.82 and 0.91, indicating good to excellent internal consistency. For specific factors, alpha values range from 0.69 to 0.77, which are considered acceptable for subscales evaluating specific symptoms [23].

**Musculoskeletal pain.** The prevalence of MSKP was estimated from an ad-hoc questionnaire with a Nordic pain questionnaire. Its psychometric properties include internal consistency, indicated by Cronbach's alpha values ranging from 0.727 to 0.816 in the Spanish version, reflecting a good level of reliability. Regarding construct validity, it was confirmed through factor analysis [24]. Participants had to answer questions about the presence or absence of pain during the last 6 months in different body regions: neck, neck and head, shoulders, upper back, and lower back. These questions had the same wording pattern and were as follows: Have you had pain or discomfort in your neck during the last 6 months? Each question was accompanied by a graphic representation of the body, along with a marking of the area indicated in the statement, to show the limits and territory of the indicated area on the body surface. When the answer was positive for pain in any region, was asked about the intensity of pain with the visual analogue scale, rating it from zero to ten.

The Musculoskeletal Pain (MSKP) index was developed to measure the severity of musculoskeletal pain reported by individuals. The index aggregates pain intensity ratings across four body regions: the neck, middle back, lower back, and shoulders. Each region's pain intensity is categorized into three levels: a rating of 1 for intensities between 1 and 3, a rating of 2 for intensities between 4 and 7, and a rating of 3 for intensities larger than 7. The MSKP index, with a maximum of 12, is derived from the sum of the pain intensity ratings across all four regions. A student is defined to be a positive

case of intense pain if their MSKP index is equal to or above 4. Since a single location can contribute up to 3 points to the MSKP index, a minimum of 4 is scored when pain is present with significant intensity in at least two locations. The proposed MSKP index assigns scores to pain intensity in four regions, with a cutoff of ≥4 indicating significant pain severity, providing a quantitative measure of pain burden. In contrast, the Nordic Musculoskeletal Questionnaire primarily captures the presence or absence of pain across regions without a standardized score or severity thresholds, focusing on prevalence and distribution rather than intensity.

### Data analysis

Table S3.1 in the S3 Appendix details the entirety of the variables measured in the survey. All categorical variables have been one-hot encoded to yield a completely numerical database. The input data has been standardized to ensure numerical stability and to facilitate the comparison between the obtained coefficients. A logistic regression model has been developed to assess the risk of the occurrence of MSKP in the surveyed population. To improve the numerical stability of the model and facilitate the comparison between the obtained coefficients, the input data has been standardized. The dimensionality of the model has been reduced by means of a two-step feature selection mechanism consisting of a correlation-driven feature screening and a Lasso-regularized logistic regression (λ=10). The final version of the logistic regression model has been trained using the selected subset of features, its performance has been evaluated in a test subset that was kept aside from the training process. Since the database is relatively small, the best set of hyperparameters for the final model has been selected using a repeated cross-validation scheme. The importance of each of the input variables in the MSKP prediction model has been evaluated with a statistical analysis on the coefficients obtained for each of the cross-validation sets. From this analysis, the average odd ratio (OR) for each variable and their corresponding 90% confidence interval (CI) are calculated, the CI have been Bonferroni corrected to account for the number of features in the final logistic regression model.

## Results

The study included 680 students, with an average age of 13.9 years (SD = 2.0). As shown in Table 1, in terms of gender distribution, 53.8% were female, while 46.1% were male. The average score for mobile dependence was 25.1 (SD = 16). Analysis of sleep quality showed that most students (41%) reported experiencing tiredness and difficulty waking up, followed by difficulty initiating sleep (35%). Regarding pain, a high percentage was reported in the middle back (30%), followed by neck pain (28.7%).

Fig 1 shows the distribution among the surveyed students of the musculoskeletal pain (MSKP) index. Under the definition of positive cases, described in the methods section, the total surveyed population is divided into two groups: negatives (526 students) and positives (154 students). The prevalence of positive cases is thus $q$ = 22.6% [95% CI: 19.7% − 25.9%].

### First feature selection step: correlation

Fig 2 shows the correlation map computed among the input variables of the model. Several correlation groups can be identified. A significant positive correlation is evident among almost all the variables related to physical activity and the home-related activities. Variables created in the one-hot encoding process from categorical variables are negatively correlated, this is the case for the group of variables related to the age group and the school transport. Notably, there is also a significant positive correlation among all the variables related to sleep problems, which suggests that individuals usually experience together different kinds of sleeping problems.

Significant correlations between input variables affect negatively the performance and stability of logistic regression models. A common approach to avoid these complications is to remove variables with large correlations. To guide the feature selection during this initial selection phase, the correlation hierarchy dendrogram (Figure S3.1 in S3 Appendix)

**Table 1. Distribution of relevant variables among the surveyed sample.**

| Variable | Distribution |
|---|---|
| Sex Female – Male | 53.8% − 46.1% |
| Age (Mean SD) | 13.9 (2.0) |
| Mobile dependence (Mean SD) | 25.1 (16.0) |
| Mobile phone dependence is Higher than 33 points | 43% (39.3-46.7) |
| Sleep Factor 1: Difficulty initiating sleep | 35.2% (31.5-38.9) |
| Sleep Factor 2 Nightmares | 13.8% (11.4-16.2) |
| Sleep Factor 3 Night awakenings | 6.1% (4.3-8.0) |
| Sleep Factor 4 Somnambulism | 9,6% (7.4-11.9) |
| Sleep Factor 5 tiredness and difficulty waking up | 41,5% (37.5-46.0) |
| Sleep Factor 6 daytime sleepiness | 29,3% (25.7-32.9) |
| IPAQ Very Low | 15% (11.7-18.3) |
| IPAQ Low | 38% (34.1-41.9) |
| IPAQ Moderate | 35% (31.6-38.4) |
| IPAQ High | 10% (7.7-12.3) |
| IPAQ Very high | 2% (0.5-3.5) |
| Neck Pain | 28.7% (24.6-32.8) |
| Shoulder Pain | 21.6% (18.0-25.2) |
| Middle back pain | 30.2% (26.1-34.4) |
| Lower back pain | 15.9% (12.7-19.1) |

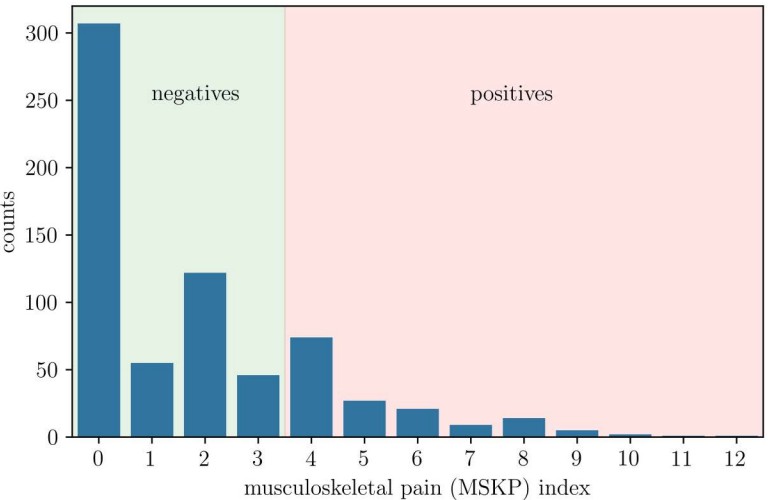

**Fig 1. Distribution of the musculoskeletal pain (MSKP) index among the surveyed sample.** Positive cases are defined as the students where the musculoskeletal pain index is equal or larger than 4.

was utilized as a guide to detect the most significant correlations. Lower connections in this dendrogram indicate a stronger Spearman correlation between variables or groups of variables. Some groups of highly correlated features can be identified, two clear examples are the one conformed by the features related to the frequency of physical activity, or the one of features related to sleep quality. Within these groups only some of the features have been selected in the next regression step. The features selected as input variables for the logistic regression model are the ones shown in blue in

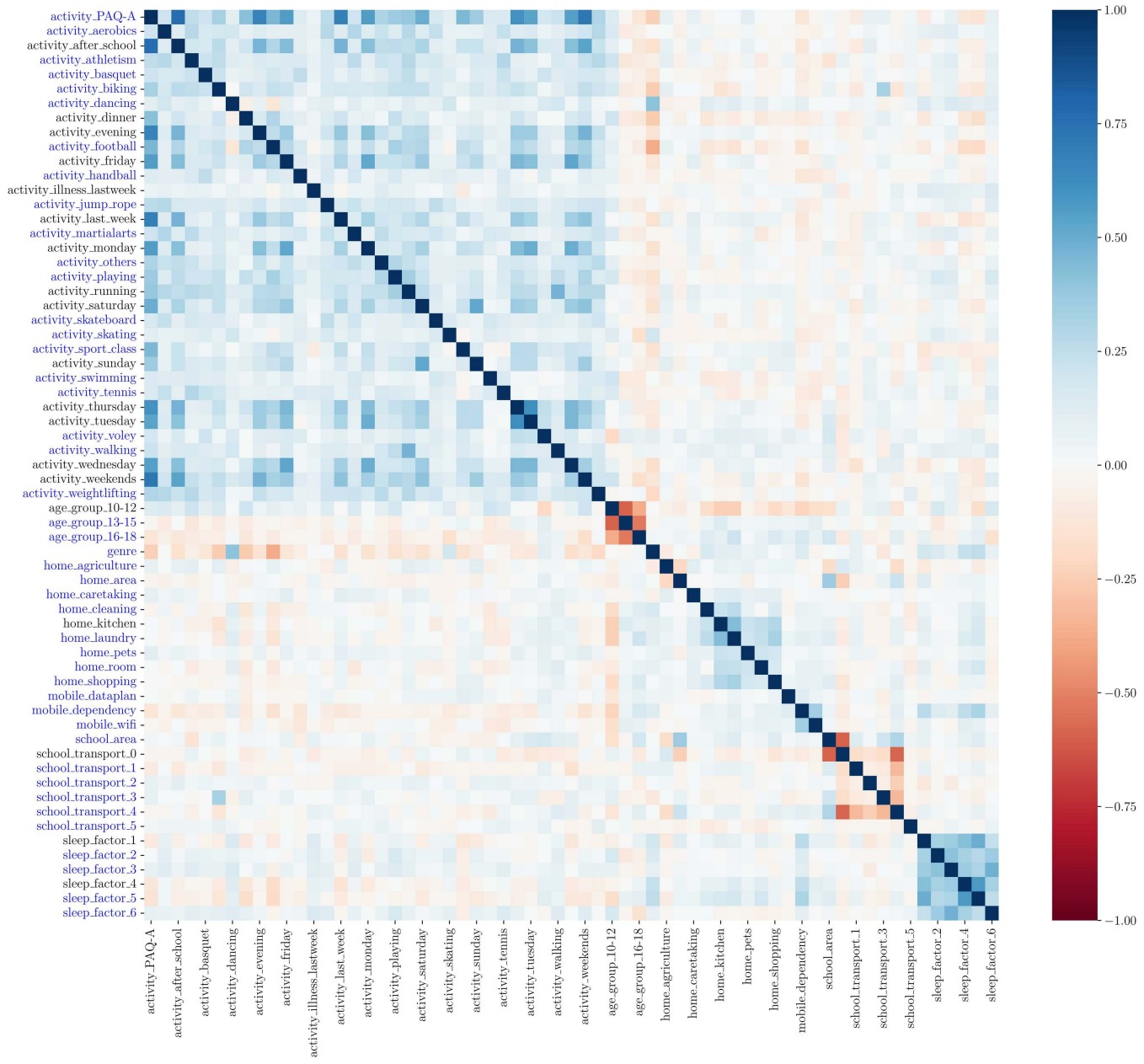

**Fig 2. Spearman correlation between all the variables obtained from the applied survey.** Variables written in blue on the y-axis are selected as input variables after the first feature selection step.

the y-axis of Fig 2. The total number of input variables for the model after the first feature selection step is thus reduced from 63 to 44.

## Second feature selection step: Lasso regularization

The next step is to build a logistic regression model with Lasso (or L1) regularization and to find the set of hyper-parameters yielding the best performance using a regular grid search. Since we are interested in maximizing the

capability of the model to distinguish between negative and positive cases, given the input data, the area under the curve (AUC) of the receiving operating characteristic (ROC) curve has been chosen as the performance metric of interest. Given the relatively small sample size, we have implemented a cross-validation scheme consisting of a 20-times repeated stratified 3-fold, yielding a total of 60 different cross-validation sets with the same prevalence of the entire sample. The best set of hyperparameters is chosen as the one yielding the best averaged AUC.

Fig 3 (a) shows the ROC curve for the best hyperparameter combination for all cross-validation sets, together with the average ROC curve. The average ROC curve is well above the chance level yielding and average AUC of 0.78(0.03), which demonstrates the capability of the model to distinguish between negative and positive cases of MSKP. The thin lines in the Fig 3 (a) show the ROC curves for each cross-validation set, a sizable variance is observed from set to set, which is the expected behavior given the small sample size and justifies the necessity of a cross-validation scheme. Fig 3 (b) shows the distribution of the odd ratio per standard deviation, obtained as the exponential of the regression coefficient, computed over all cross-validation sets. Because of the Lasso regularization, most of the variables exhibit predominantly zero coefficients, which results in an odd ratio of 1. These variables do not significantly contribute to the model and can thus be discarded in further analysis. Based on the results of Fig 3 (b), in the second feature selection step we have kept only variables with a distribution median different than zero, thus further reducing the number of features to 13.

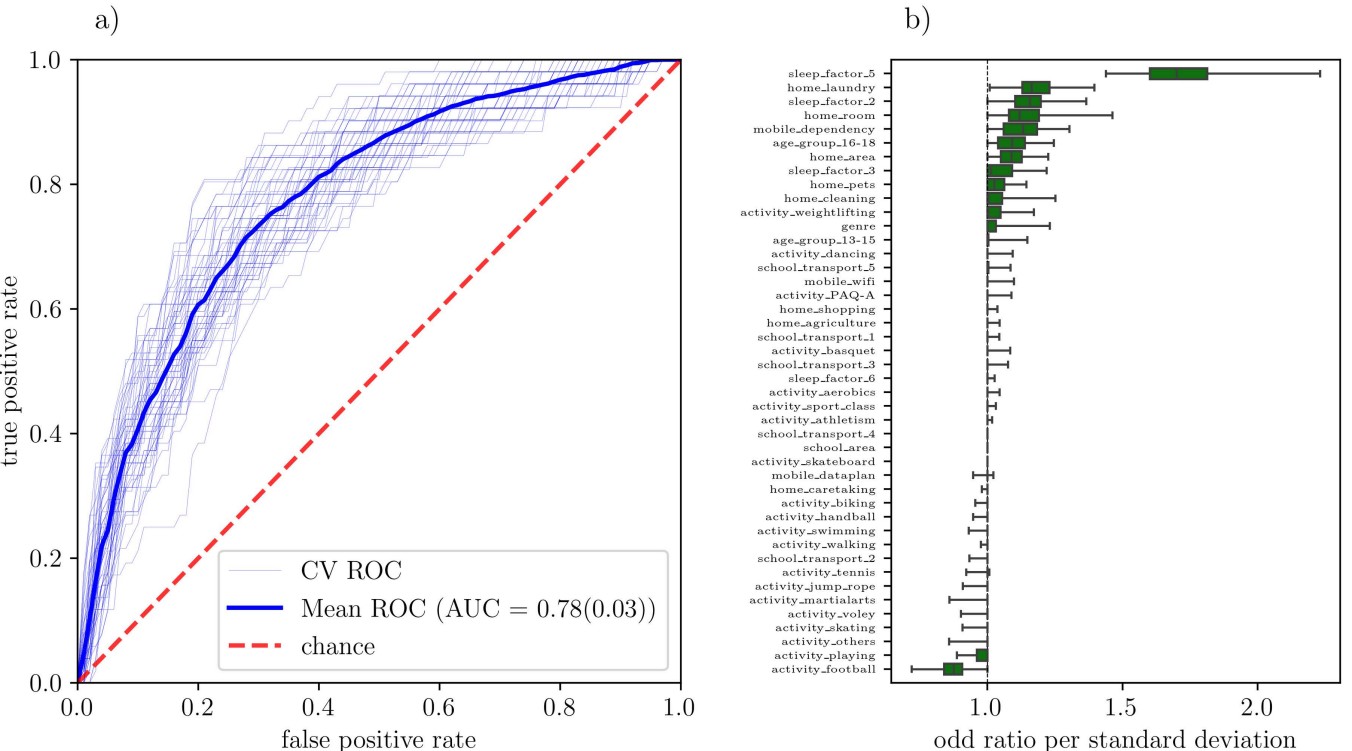

**Fig 3. Lasso-regularized logistic regression model analysis.** a) Thin lines: ROC curve for all cross-validation sets. Thick line: average ROC curve over all cross-validation sets. Dashed line: the chance baseline, shown as a reference. b) Distribution of the odd ratio per standard deviation for all input variables in the model. The distribution is computed among all 60 different train-test sets in the cross-validation. The whiskers in this case represent the maximum and minimum values of the distribution. Variables are deemed relevant if the median of the distribution is different from zero.

## Definitive logistic regression model

We now proceed to build the definitive logistic regression model. To mitigate overfitting, we have implemented the data workflow shown in Fig 4. In brief, 20% of the data have been kept aside as a risk evaluation dataset, this dataset will only be used at the end of the process to evaluate the performance of the proposed risk model. The remaining data has been divided into a training subset (60% of the sample) and a risk definition subset (20% of the sample). The latter is used to define the risk thresholds and expected positive prevalence for different risk levels. The prevalence of positive cases in all subsets is preserved equal to the prevalence of the entire sample. The logistic regression model has been trained on the train subset using, once again, a cross-validation scheme consisting of a 20-times repeated stratified 3-fold. The best hyperparameters are the ones yielding the best average AUC of the ROC curve.

The definitive model is defined as an average over the ensemble of all the cross-validation fitted models using the hyperparameters yielding the best performance on the train set. To evaluate the performance of the model on unknown data, each of the cross-validation models has been applied to the risk definition subset. The results are shown in Fig 5

**Fig 4. Final model data workflow.** The original dataset is split into a Risk Evaluation set (20% of samples) and an intermediate dataset (80%). The intermediate dataset is split into the model training dataset (60% of total sample) and a risk determination dataset (20% of total sample). The model training dataset is used to train 60 cross-validation models. The results of these models are used on the risk determination dataset to define the risk levels and score thresholds. Finally, the risk evaluation dataset is used to evaluate and compare the observed prevalence on the different risk levels.

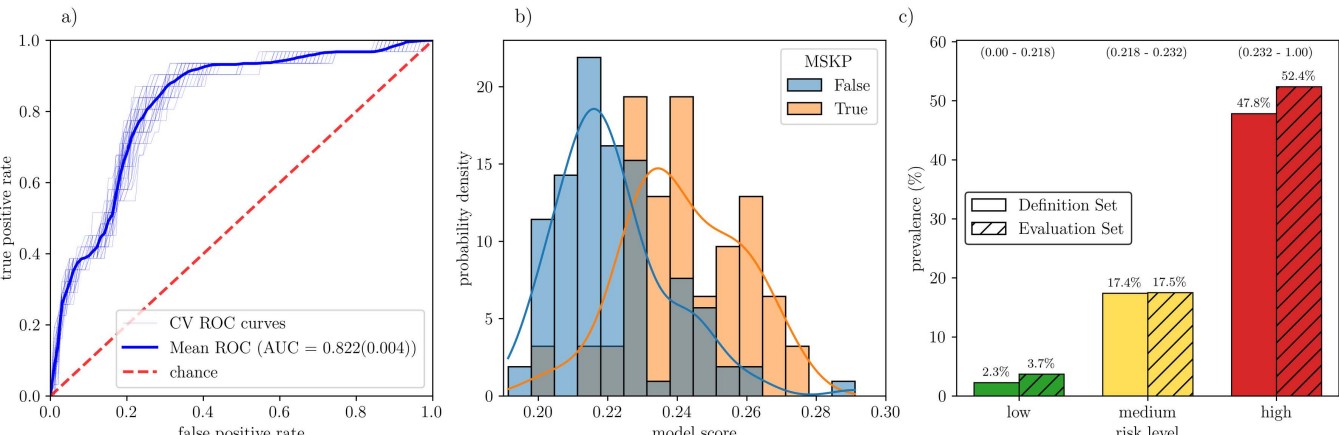

**Fig 5. Final model analysis.** a) Thin lines: ROC curve evaluated over the risk evaluation set for each of the cross-validation models. Solid line: ROC curve over the risk evaluation set obtained by averaging the score obtained with each of the fitted models in the cross-validation. Dashed line: the chance baseline, shown as a reference. b) Model score distribution for negative and positive cases in the risk evaluation set. c) Comparison of the prevalence of positive cases between the risk definition and the risk evaluation test sets. The risk group is assigned using the score ranges depicted on the upper part of the figure.

(a) where the ROC curves produced by each cross-validation model and their average are shown. The average AUC is 0.822(0.004), which shows that the model does not lose performance despite working with only 13 features.

Fig 5 (b) shows the distribution of the model score over all negative and positive students in the risk definition set. The model score is computed as the average of the score obtained for each of the cross-validation models. As can be seen, the distribution for the positive cases is slightly shifted towards higher scores, which suggests that subjects yielding lower scores are more likely to be negative cases, while subjects yielding higher scores are more likely to be positive.

Using these scores, we propose the risk level assessment shown in Fig 5 (c), model score tertiles have been used to define score ranges and the risk of MSKP has been estimated as the fraction of positive cases in the risk definition set. The risk levels are thus defined as follows: for subjects with an average score below 0.218 a MSKP prevalence of 2.3% is expected, which is roughly 10 times lower than the overall prevalence, this group is assigned a low MSKP risk. For subjects with an average score between 0.218 and 0.232 a MSKP prevalence of 17.4% is expected, which is comparable to the sample prevalence, this group is assigned a medium MSKP risk. For subjects with an average score above 0.232 a MSKP prevalence of 47.8% is expected, which is more than twice the sample prevalence, this group is assigned a high MSKP risk. Since the score ranges have been defined using tertiles, it is expected that the three risk groups are equally represented in any sample.

To test the capability of the model to assess the MSKP risk on unseen data, we have evaluated the model on the risk evaluation subset. The results are also shown in Fig 5 (c), which shows that the prevalence of MSKP in the low, medium and high-risk groups is 3.7%, 17.5% and 52.4% respectively. These values are similar to the prevalence in the risk definition set and thus testify to the capability of our model to assess the MSKP risk on unseen data.

An additional advantage of the logistic regression models is their interpretability; the computed coefficients of each input variable can directly be related to its relevance in the computation of the risk of MSKP. More specifically, the coefficient corresponds to the natural logarithm of the odd ratio of developing MSKP when the value of the input variable increases by one standard deviation [25]. For each cross-validation set, we have obtained the odd ratio per standard deviation by exponentiating the coefficients, Fig 6(a) shows the resulting distribution among all cross-validation sets of the odd ratio for each of the 13 selected input features.

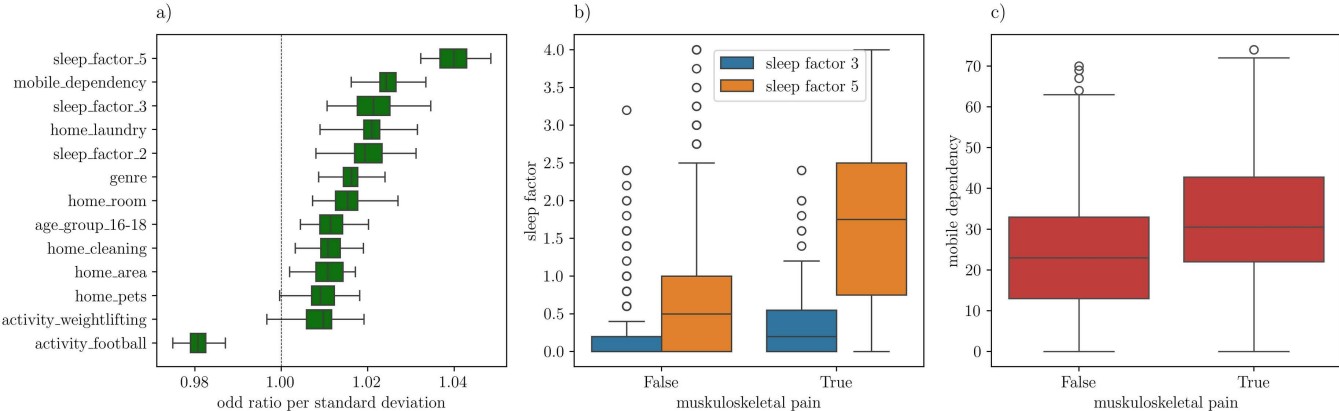

**Fig 6. a) Distribution of the odd ratio per standard deviation for all input variables in the final logistic regression model.** The distribution is computed among all 60 different train-test sets in the cross-validation. The whiskers in this case represent the maximum and minimum values of the distribution. b) Distribution of the features sleep_factor_3 and sleep_factor_6 for negative and positive cases of musculoskeletal pain. C) Distribution of the mobile dependency score for negative and positive cases of musculoskeletal pain.

The most relevant variable is *sleep_factor_5*, related to tiredness and difficulty waking up, as specified in Table A. 1 in the Supplementary Material. The average value of the coefficient for this variable is 0.039, which results in an average odd ratio of , while the standard deviation of the variable in the surveyed sample is 1.04. The results can therefore be interpreted as follows: on average, for the participants of this study an increment of 1.04 in *the sleep_factor_5* results in a 4% higher probability of experiencing MSKP. Since the *sleep_factor_5* ranges from 0 to 4, the odd ratio for the whole range can be estimated to be . This is, a student with a *sleep_factor_5* of 4, is 16.3% more likely to experience MSKP than a student with a *sleep_factor_5* of 0.

Table 2 details the maximum odd ratio and its 90% confidence interval for all the selected input variables. The maximum odd ratio is defined as the odd ratio between the maximum and minimum possible values of the input variable. The confidence interval has been calculated using the statistics gathered from all 60 different cross-validation sets, it is Bonferroni corrected to consider the number of features used in the analysis. The largest maximum odd ratios are obtained for

**Table 2. Range, maximum odd ratio and 90% confidence interval for the maximum odd ratio for the relevant parameters of the model. The maximum odd ratio is defined as the odd ratio between the minimum and maximum values of the parameter. The confidence intervals have been Bonferroni corrected to account for the 13 different features in the model.**

| Parameter | Range | Maximum Odd Ratio | 90% Confidence Interval for the Maximum Odd Ratio |
|---|---|---|---|
| sleep_factor_3 | 0.0-3.2 | 1.182 | [1.086-1.302] |
| sleep_factor_5 | 0.0-4.0 | 1.163 | [1.131-1.199] |
| mobile_dependency | 0-74 | 1.117 | [1.077-1.161] |
| sleep_factor_2 | 0.0-3.0 | 1.107 | [1.044-1.167] |
| home_laundry | 0-1 | 1.044 | [1.02-1.065] |
| activity_weightlifting | 1-5 | 1.041 | [0.99-1.09] |
| home_room | 0-1 | 1.040 | [1.02-1.068] |
| genre | 0-1 | 1.032 | [1.018-1.048] |
| home_cleaning | 0-1 | 1.031 | [1.01-1.052] |
| age_group_16–18 | 0.0-1.0 | 1.026 | [1.01-1.047] |
| home_area | 0-1 | 1.022 | [1.004-1.036] |
| home_pets | 0-1 | 1.020 | [1.0-1.037] |
| activity_football | 1-5 | 0.943 | [0.927-0.962] |

variables *sleep_factor_3* and *sleep_factor_5*. The *sleep_factor_2*, on the other hand, appears in the fourth place. This is strong evidence that sleeping problems and MSKP are often related, our analysis does not provide evidence regarding the causal relation between these two problems. Fig 6(b) shows the distribution of the features *sleep_factor_3* and *sleep_factor_5* for negative and positive cases of MSKP, the difference in the distributions in both cases is sizable and confirms that there is a strong correlation.

Notably, the third variable in importance (deemed by the maximum odd ratio) is *mobile_dependency*, for which an average maximum odd ratio of 1.117 is observed. This result suggests that a student scoring 74 in the mobile dependency test is, on average, 11.7% more likely to experience MSKP than a student scoring 0. This is a remarkable result of this research, a link between the MSKP and the excessive use of mobile devices is demonstrated. Fig 6(c) confirms a sizable correlation between the mobile dependency score, *mobile_dependency*, and the occurrence of MSKP. Although the causal relationship between the use of mobile devices and the appearance of MSKP could be inferred, further research is needed to demonstrate it.

The home activities also seem to have a significant correlation with chronic pain, the input variables *home_laundry* and *home_room* are also among the most relevant to the model. As shown in Table 2, students with a 1 in these variables are on average 4% more likely to experience MSKP. In this case, however, it is natural to establish a causal relationship and to conclude that students with home responsibilities are more likely to experience intense chronic pain.

The odd ratio for the variable *age_group_16–18* in this case must be understood as the increase in the probability of being classified as positive with respect to the students in the *age_group_10–12*. The latter variable has been eliminated from our analysis, and it constitutes the baseline of the logistic regression. Students between 16 and 19 years old are, on average, 2.6% more likely to experience MSKP than younger surveyed students. The *home_area* is valued as 0 if the student resides in a rural area and 1 if the student resides in an urban area. We find that students in urban areas are on average 2.2% more likely to experience MSKP.

Among all the surveyed physical activity parameters, only the *activity_weighlifting* shows a positive correlation with the occurrence of MSKP. This observation might be related to the risk of the weightlifting malpractice. The list of features increasing the risk of MSKP is closed by *home_cleaning*, which measures if a student has cleaning responsibilities at home, *home_pets*, which measures if a student oversees pets, and *genre*, which is 0 for boys and 1 for girls. The results on the latter variable suggest that women are in higher risk of experiencing MSKP, this is probably related with the increase prevalence of MSKP among girls (28.9%) as compared to boys (15.2%).

Finally, there is only one relevant variable with a negative impact on the occurrence of MSKP: *activity_football*. With a maximum odd ratio of 0.943, we find that the students who practice football more often are on average 5.7% less likely to experience MSKP. Once again, further studies are required to establish a causal relationship between the risk of MSKP and the football practice.

## Discussion

Our research reveals that musculoskeletal pain in adolescents is linked to various factors, including sleep quality, mobile device dependency, age, physical activity, home activities, and sex. Moreover, our study introduces an MSKP index, incorporating pain intensity across four body regions. This innovative approach offers a novel method for screening musculoskeletal pain in children and adolescents. While mobile dependency was identified as a relevant predictor, its contribution (OR=1.117) should be interpreted alongside other more robust factors, such as sleep quality, which has demonstrated a more consistent and stronger relationship with MSKP in previous studies.

In terms of sleep quality, various studies have highlighted the connection between musculoskeletal pain and sleep. Scarobottolo [26] conducted research in Brazil, revealing that poor sleep quality is associated with increased likelihood of experiencing low back pain and neck pain among both boys and girls, even after accounting for factors such as age, socioeconomic status, and physical activity levels. Additionally, another study [27] found that children who spent less time

in front of screens tended to have better sleep quality than those with high screen time. Excessive screen time has been shown to impair sleep quality, leading to various adverse physical and psychological consequences [28].

Notably, except for dancing, female sex is associated negatively with all the variables related to physical activity, which suggests that surveyed girls, on average, perform less physical activity than surveyed boys. On the other hand, the female sex is positively associated with poor sleep quality indices and with home activity variables. Surveyed girls thus experience more sleeping problems and have more responsibilities at home. A study conducted in Saudi Arabia involving 2000 adolescents yielded results consistent with those of the present research. It revealed that the occurrence of pain was linked to factors such as sex, age (particularly around 16 years old), sleep quality, and insufficient physical activity [29].

Regarding mobile dependence our study finds that the *mobile_dependecy*, which measures the student's dependence to the mobile phone, comes in third place as a relevant input variable, this is the only variable related to mobile phone usage deemed relevant, as the variables related to the availability of Wi-Fi (*mobile_wifi*) or a data plan (*mobile_dataplan*) are not relevant.

According to our analysis, a student with the maximum score in TDM is on average 11.7% more likely to be predicted positive for intense pain. This is a significant finding as it evidences a direct relationship between the presence of intense chronic pain and the excessive usage of mobile phones among the surveyed students. It is related to the results of Gorce et al. [30] who observed that students who exhibited more problematic phone usage tended to adopt various awkward postures, particularly reclining postures, especially during evening hours. These postures were identified as potentially contributing to the development of Musculoskeletal Disorders over the medium to long term.

All variables about the frequency and timing of physical activity demonstrate an association with the PAQ-A score (*activity_PAQ-A*), which is a direct consequence of the score's definition. Furthermore, our findings indicate an association between the age group at major risk for Musculoskeletal (MKS) pain and the age range of 16–18 years. This association may be attributed to mobile device dependence coupled with low physical activity levels, as it has been reported that the frequency of physical activity notably declines as children progress into older age groups [31]. The protective effect of football may be related to improvements in overall physical fitness, coordination, and muscle strength, which contribute to a reduced vulnerability to musculoskeletal pain.

The proposed MSKP risk assessment allows the segmentation of the student population into different risk categories and the identification of students at high risk of developing MSKP. The MSKP index is a novel approach that, unlike the Nordic Musculoskeletal Questionnaire, not only assesses the presence but also the intensity of pain in different regions, allowing for a more precise risk categorization tailored to the school population.

## Conclusion

MSKP is a validated tool designed for screening adolescents at high risk of developing musculoskeletal pain. Key factors associated with increased risk include excessive mobile device usage, disrupted sleep patterns, engagement in household chores (such as laundry and bedroom maintenance), being 16–18 years old, and residing in urban areas. Regular participation in physical activity, particularly football, was identified as a factor negatively correlated to the occurrence of MSKP. Adolescents who engaged frequently in football activities were found to be 5.7% less likely to develop musculoskeletal symptoms. However, given the cross-sectional nature of the study, these associations do not imply causality, and longitudinal studies are needed for validation. The risk scale derived from our model facilitates the identification of individuals at higher risk, which can enhance targeting of future prevention and intervention efforts.

## Limitations

One of the study's limitations is that confounding factors such as nutrition, mental health, and socioeconomic status were not considered, yet they may influence the risk of MSKP. Additionally, the case definition (MSKP score ≥4) requires external validation to confirm its clinical utility and predictive capacity. The non-probabilistic sampling, lack of validation in other

populations, and potential overfitting of the model also limit the generalizability of our findings. To strengthen external validity and the robustness of the MSKP index, future research should include more diverse samples and employ multi-center validation processes.

## Declaration of generative AI and AI-assisted technologies in the writing process

While preparing this work, the authors used chatGPT 3.5 to correct style and grammar mistakes. After using this tool/service, the authors reviewed and edited the content as needed and took full responsibility for the of the published article.

## Supporting information

**S1 Questionnaire. Questionnaire Spanish.** Original applied questionnaire, in spanish.
(PDF)

**S2 Questionnaire. Questionnaire English.** English translation of the applied questionnaire.
(PDF)

**S3 Appendix. Appendix.** Contains the detail of the variables considered in the model (Table S3.1), and the correlation dendrogram (Figure S3.1).
(PDF)

## Acknowledgments

The authors would like to thank the public-school authorities and participants of this study.

## Author contributions

**Conceptualization:** Margareth Lorena Alfonso-Mora, María Alejandra Sanchez Vera, Andrea Milena García Becerra, Paola Sarmiento Gonzalez.

**Data curation:** Miguel Angel Uribe-Laverde.

**Formal analysis:** Miguel Angel Uribe-Laverde.

**Methodology:** Margareth Lorena Alfonso-Mora, María Alejandra Sanchez Vera, Andrea Milena García Becerra.

**Project administration:** Margareth Lorena Alfonso-Mora.

**Resources:** Paola Sarmiento Gonzalez.

**Supervision:** Margareth Lorena Alfonso-Mora, María Alejandra Sanchez Vera.

**Writing – original draft:** Miguel Angel Uribe-Laverde, Margareth Lorena Alfonso-Mora.

**Writing – review & editing:** Miguel Angel Uribe-Laverde, María Alejandra Sanchez Vera, Andrea Milena García Becerra, Paola Sarmiento Gonzalez.

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
