## [Decision Letter · Decision Letter 0]

17 Mar 2025

PONE-D-24-55797MSKP INDEX: RISK MODEL OF MUSCULOSKELETAL PAIN IN A COLOMBIAN ADOLESCENT SAMPLEPLOS ONE

Dear Dr. Uribe-Laverde,

Thank you for submitting your manuscript to PLOS ONE. After careful consideration, we feel that it has merit but does not fully meet PLOS ONE’s publication criteria as it currently stands. Therefore, we invite you to submit a revised version of the manuscript that addresses the points raised during the review process.

We look forward to receiving your revised manuscript.

Kind regards,

Alejandro Botero Carvajal, MD

Academic Editor

PLOS ONE

Journal Requirements:

 “This study was funded by the University of La Sabana (2020 – Convocatoria Interna Para La Financiación De Proyectos De Investigación, Creación, Desarrollo Tecnológico E Innovación). Project Id: ENF-59-2020.” 

4. Please note that your Data Availability Statement is currently missing the repository name and/or the DOI/accession number of each dataset OR a direct link to access each database. If your manuscript is accepted for publication, you will be asked to provide these details on a very short timeline. We therefore suggest that you provide this information now, though we will not hold up the peer review process if you are unable.

Reviewers' comments:

Reviewer's Responses to Questions

**Comments to the Author**

1. Is the manuscript technically sound, and do the data support the conclusions?

Reviewer #1: No

Reviewer #2: Yes

2. Has the statistical analysis been performed appropriately and rigorously? 

Reviewer #1: Yes

Reviewer #2: Yes

3. Have the authors made all data underlying the findings in their manuscript fully available?

Reviewer #1: Yes

Reviewer #2: Yes

4. Is the manuscript presented in an intelligible fashion and written in standard English?

Reviewer #1: Yes

Reviewer #2: Yes

5. Review Comments to the Author

Reviewer #1: Authors and Editor, Dear

This manuscript, "MSKP INDEX: RISK MODEL OF MUSCULOSKELETAL PAIN IN A COLOMBIAN ADOLESCENT SAMPLE," presents an intriguing effort to develop a logistic regression-based risk model for musculoskeletal pain (MSKP) among Colombian adolescents. The study leverages a sizable sample (n=680) and addresses a regionally underexplored issue, proposing a novel MSKP index and risk scale with potential practical applications. The use of validated tools and advanced statistical methods (e.g., Lasso regularization, cross-validation) demonstrates methodological ambition. However, the manuscript requires significant revisions to meet the rigorous standards of a JCR-indexed journal like PLOS ONE. Key concerns include methodological ambiguities (e.g., MSKP index validation, feature selection details), statistical shortcomings (e.g., lack of overfitting checks, unadjusted odds ratios), and interpretive overreach (e.g., implied causality in a cross-sectional design). Below, I outline the main criticisms for each section, offering specific recommendations to enhance clarity, rigor, and impact. I recommend major revisions to address these issues, confident that the study’s potential can be realized with careful refinement.

Title and Keywords

Revise the title to eliminate redundancy (e.g., "Development of the MSKP Index: A Risk Model for Musculoskeletal Pain in Colombian Adolescents") and consider adding methodological specificity (e.g., "Logistic Regression Model").

Replace "Physiotherapy" in keywords with "Logistic Regression" or "Risk Assessment" to align with the study’s core methodology.

Abstract

Specify the MSKP prevalence (e.g., "22.6% of participants reported MSKP") to enhance informativeness.

Detail the methods by briefly mentioning feature selection (e.g., "using correlation analysis and Lasso regularization") and validation (e.g., "cross-validated on a test set").

Trim the abstract to fit PLOS ONE’s 200–250-word limit while retaining essential details, ensuring conciseness without sacrificing clarity.

Introduction

Expand the literature review to include a broader geographical and methodological scope (e.g., European studies like Hestbaek et al., 2006, Spine).

Use cautious language to avoid unsupported causal claims (e.g., "may contribute to" instead of "trigger" for posture and pain).

Substantiate the Latin American research gap with evidence (e.g., cite a review or meta-analysis).

Articulate a specific hypothesis (e.g., "We hypothesized that mobile dependency and sleep quality would be significant predictors of MSKP") to provide a theoretical foundation.

Methods

Address potential seasonal effects of data collection (September–October 2022) on physical activity or pain reporting.

Recalculate sample size using a realistic prevalence estimate (e.g., 20–30% from prior studies) instead of the unjustified 50%, and report a power analysis.

Acknowledge non-probabilistic sampling as a limitation and clarify vague exclusion criteria (e.g., specify types of "diagnosed disabilities").

Validate the MSKP index cutoff (≥4) against clinical standards (e.g., pain duration, functional impairment) and justify pain intensity categories (1–3, 4–7, >7) with psychometric evidence.

Provide reliability statistics for all instruments (e.g., IPAQ, sleep scale), not just TDM.

Detail feature selection by specifying the correlation threshold and Lasso regularization parameters (e.g., lambda).

Justify the cross-validation scheme (20x 3-fold) relative to sample size and clarify the standardization method (e.g., z-scoring).

Adjust odds ratios (ORs) for multiple comparisons (e.g., Bonferroni) to reduce type I error risk.

Results

Add confidence intervals or p-values to Table 1 for prevalence estimates and contextualize the mobile dependency score (e.g., clinical cutoff).

Discuss the correlation dendrogram (Figure A.1) in the text to guide interpretation.

Benchmark the Lasso model AUC (0.78) against simpler models (e.g., univariate regression) and quantify ROC curve variance (e.g., SD of AUC).

Assess overfitting risk in the final model (e.g., via calibration plots or Brier score) and report sensitivity/specificity for each risk scale cutoff (Figure 4c).

Test linearity assumptions for ORs (e.g., sleep_factor_5 OR = 7.89) and discuss coefficient instability indicated by wide CIs (e.g., [4.72–11.9]).

Discussion

Use precise language (e.g., "associated with" instead of "correlated") to reflect logistic regression results accurately.

Balance emphasis on mobile dependency (OR = 1.72) relative to stronger predictors like sleep factors, avoiding overhyping its role.

Hypothesize mechanisms for football’s protective effect (OR = 0.66), such as aerobic fitness or coordination, to deepen interpretation.

Engage critically with prior studies (e.g., explain differences from Thailand/Japan findings) rather than offering superficial comparisons.

Benchmark the MSKP index against existing tools (e.g., Nordic Musculoskeletal Questionnaire) to substantiate novelty claims.

Conclusion

Specify the MSKP index’s intended application (e.g., "for screening high-risk adolescents") rather than using vague terms like "promising."

Avoid causal language (e.g., "contributing to this risk") given the cross-sectional design, emphasizing the need for longitudinal validation.

Limitations

Acknowledge additional limitations: non-probabilistic sampling, lack of external validation, and potential overfitting risk.

Clarify the ambiguous suggestion of "multiple regression models" (e.g., specify multinomial logistic regression or another approach).

Additional Sections

Complete funding details per PLOS ONE guidelines (e.g., include grant amount for Project ID: ENF-59-2020).

Resolve the placeholder Harvard Dataverse DOI ("XXX") with a functional link.

Update references to include key predictive modeling studies (e.g., Hestbaek et al., 2006; Dunn et al., 2013) and address timeline issues (e.g., remove or justify 2024 citations post-dating 2022 data collection).

Reviewer #2: Peer Review for Manuscript PONE-D-24-55797: MSKP INDEX: RISK MODEL OF MUSCULOSKELETAL PAIN IN A COLOMBIAN ADOLESCENT SAMPLE

General Assessment

The study presents a risk model for musculoskeletal pain (MSKP) in Colombian adolescents, addressing an important topic. However, several issues related to methodology, data presentation, and interpretation need to be addressed for clarity and scientific rigor.

Major Comments

1. Introduction

• The study lacks a clear theoretical framework explaining how risk factors (e.g., mobile dependency, sleep disorders) contribute to MSKP.

• The global prevalence of adolescent MSKP should be briefly discussed to highlight the study’s significance.

2. Methodology

• The broad age range (10–18 years) is not justified. Adolescents experience different developmental changes, so age-stratified analysis is recommended.

• The case definition (MSKP score ≥4) needs validation. Why was this threshold chosen?

• The feature selection method (Lasso Regression) should be justified against alternatives like PCA or stepwise regression.

• Confounding factors (nutrition, mental health, socioeconomic status) are not considered but could influence MSKP risk.

3. Results

• The 22.6% prevalence of MSKP should be reported with confidence intervals for accuracy.

• Regression coefficients should include effect sizes and confidence intervals to assess the strength of associations.

• The logistic regression model (AUC = 0.83) is promising but needs calibration plots and sensitivity/specificity values to confirm robustness.

4. Discussion

• The study suggests causal relationships, but given its cross-sectional design, only associations can be inferred.

• The sample is region-specific (Colombia), limiting generalizability. This should be acknowledged.

• A stronger comparison with existing research would help contextualize the findings.

5. Limitations

The limitations section should be included, highlighting:

• The cross-sectional nature (no causal conclusions).

• The lack of objective measures for physical activity and sleep quality.

Minor Revisions

• Abstract: Should briefly mention key limitations (e.g., cross-sectional design, self-reported data).

• Figures & Tables:

o Add scatterplots for key correlations (e.g., mobile dependency vs. MSKP risk).

o Ensure confidence intervals are included in tables.

• References: Ensure citations follow PLOS One formatting and include key MSKP studies.

6. PLOS authors have the option to publish the peer review history of their article (what does this mean? ). If published, this will include your full peer review and any attached files.

**Do you want your identity to be public for this peer review?** For information about this choice, including consent withdrawal, please see our Privacy Policy .

Reviewer #1: No

Reviewer #2: **Yes: ** Cihan Aygün

---

## [Author Response · Author response to Decision Letter 1]

3 Jul 2025

We have uploaded a rebutal letter with all the responses to the reviewer's comments.

---

## [Decision Letter · Decision Letter 1]

3 Aug 2025

DEVELOPMENT OF THE MSKP INDEX: RISK MODEL OF MUSCULOSKELETAL PAIN IN COLOMBIAN ADOLESCENTS

PONE-D-24-55797R1

Dear Dr. Uribe-Laverde,

We’re pleased to inform you that your manuscript has been judged scientifically suitable for publication and will be formally accepted for publication once it meets all outstanding technical requirements.

Kind regards,

Alejandro Botero Carvajal, Ph.D

Academic Editor

PLOS ONE

Additional Editor Comments (optional):

Reviewers' comments:

Reviewer's Responses to Questions

**Comments to the Author**

1. If the authors have adequately addressed your comments raised in a previous round of review and you feel that this manuscript is now acceptable for publication, you may indicate that here to bypass the “Comments to the Author” section, enter your conflict of interest statement in the “Confidential to Editor” section, and submit your "Accept" recommendation.

Reviewer #1: All comments have been addressed

Reviewer #2: All comments have been addressed

2. Is the manuscript technically sound, and do the data support the conclusions?

Reviewer #1: Yes

Reviewer #2: Yes

3. Has the statistical analysis been performed appropriately and rigorously? 

Reviewer #1: Yes

Reviewer #2: Yes

4. Have the authors made all data underlying the findings in their manuscript fully available?

Reviewer #1: Yes

Reviewer #2: Yes

5. Is the manuscript presented in an intelligible fashion and written in standard English?

Reviewer #1: Yes

Reviewer #2: Yes

6. Review Comments to the Author

Reviewer #1: (No Response)

Reviewer #2: The authors have comprehensively addressed reviewer comments with appropriate methodological enhancements and a clearer presentation. The revisions demonstrate good scientific practice and significantly strengthen the manuscript's quality.

7. PLOS authors have the option to publish the peer review history of their article (what does this mean? ). If published, this will include your full peer review and any attached files.

**Do you want your identity to be public for this peer review?** For information about this choice, including consent withdrawal, please see our Privacy Policy .

Reviewer #1: No

Reviewer #2: **Yes: ** Cihan Aygün

---

## [Editor Report · Acceptance letter]

PONE-D-24-55797R1

PLOS ONE

Dear Dr. Uribe-Laverde,

I'm pleased to inform you that your manuscript has been deemed suitable for publication in PLOS ONE. Congratulations! Your manuscript is now being handed over to our production team.

Kind regards,

on behalf of

Dr. Alejandro Botero Carvajal

Academic Editor

PLOS ONE